# Nutraceutical COMP-4 confers protection against endothelial dysfunction through the eNOS/iNOS-NO-cGMP pathway

Monica G. Ferrini[1], Andrea Abraham[1], Revecca Millán[1], Leslie Graciano[1], Sriram V. Eleswarapu[2]*, Jacob Rajfer[2]

1 Department of Health and Life Sciences, Charles R. Drew University of Medicine and Science, Los Angeles, California, United States of America, 2 Department of Urology, David Geffen School of Medicine at UCLA, Los Angeles, California, United States of America

* seleswarapu@mednet.ucla.edu

**Data Availability Statement:** All relevant data are within the manuscript and its Supporting Information files (ZIP file with all raw data that constitute all figures).

## Abstract

The nutraceutical COMP-4 –consisting of L-citrulline, ginger extract, and herbal components *Paullinia cupana* and muira puama–has been shown previously to stimulate the production of nitric oxide (NO) in a variety of tissue types. We hypothesized that COMP-4 may have a protective, stimulatory effect on the vascular endothelial cell. Human umbilical arterial endothelial cells were incubated for 24 hours with or without COMP-4 and, to replicate impairment of endothelial function, co-incubated with or without $H_2O_2$. NO intracellular content, nitrite formation and cGMP content in culture media, nitric oxide synthase (NOS) isoforms and mRNA content, pro-inflammatory cytokines, and PAI-1 expression and activity were measured. COMP-4 increased endothelial cell production of NO and cGMP and the expression of both endothelial NOS (eNOS) and inducible NOS (iNOS), in tandem with a reduction in cytokine expression and activity of PAI-1. Co-incubation of COMP-4 with $H_2O_2$ reversed detrimental effects of $H_2O_2$ on endothelial function, evidenced by improvement in NO availability and abrogation of the pro-inflammatory milieu. These results suggest that COMP-4 exerts a stimulatory effect on endothelial cell eNOS and iNOS to increase NO bioavailability, leading to a reduction in pro-inflammatory cytokines, particularly the prothrombotic PAI-1.

## Introduction

The vascular endothelium is an essential modulator of cardiovascular function [1]. Endothelial cells contribute to the response of blood vessels to different physiological and pathological stimuli. Abnormalities in endothelial cell structure and function may lead to cardiovascular diseases [1, 2].

Nitric oxide (NO) is not only an essential mediator in regulating endothelial cell function but is also synthesized by the endothelial cell itself [3]. NO regulates numerous biochemical activities within endothelial cells, such as maintenance of endothelial integrity [3], reduction of platelet deposition and aggregation [4], decrease in leukocyte adhesion [5], inhibition of

**Funding:** This work was supported by a grant from the Peter A. Morton Foundation; and the NIH-National Institute on Minority Health and Disparities (NIH/NIMHD) [grant number U54MD007598 to MGF]. The funders had no role in study design, data collection and analysis, decision to publish, or preparation of the manuscript.

**Competing interests:** JR is a principal in KLRM, LLC, which holds the patent for COMP-4. All other authors have declared that no competing interests exist. This does not alter our adherence to PLOS ONE policies on sharing data and materials.

**Abbreviations:** cGMP, cycling guanosine monophosphate; CXCL-1, chemokine (C-X-C) ligand-1; DAF-FM DA, 4-amino-5-methylamino-2',7'-difluorofluorescein diacetate; DETA NONOate, diethylenetriamine/nitric oxide complex sodium salt hydrate; eNOS, endothelial nitric oxide synthase; G, ginger; H$_2$O$_2$, hydrogen peroxide; HUAEC, human umbilical arterial endothelial cells; IBMX, 3-isobutyl-1-methylxanthine; IL-6, interleukin-6; IL-8, interleukin-8; iNOS, inducible nitric oxide synthase; L-NIL, L-N6-(1-iminoethyl)lysine; L-NAME, L-nitro arginine methyl ester; LPS, lipopolysaccharide; MIF, macrophage migration inhibitory factor; MP, muira puama; nNOS, neuronal nitric oxide synthase; NO, nitric oxide; PAI-1, plasminogen activation inhibitor-1; PC, Paullinia cupana; ROS, reactive oxygen species; tPA, tissue plasminogen activator.

smooth muscle cell proliferation and migration [6], and induction of vasodilation [7]. A reduced NO bioavailability at the endothelial cell layer contributes to endothelial dysfunction [8], which enhances the development of vascular fibrosis and atherosclerosis [9, 10].

It is assumed that the constitutively expressed endothelial nitric oxide synthase (eNOS or NOS3) is responsible for maintaining and improving endothelial function during chronic inflammatory conditions such as atherosclerosis or stenosis [9, 11]. However, under chronic pro-inflammatory conditions, a local expression of the inducible NOS (iNOS or NOS2) isotype is seen in endothelial and other cell types [12]. Activation of iNOS leads to high-output NO synthesis, which was initially perceived as a response associated with local tissue destruction and cell death [12]. However, more recent investigations have demonstrated a powerful protective effect of NO derived from iNOS toward cellular stress conditions [13]. Moreover, it has been shown that iNOS-derived NO modulates the expression of many different genes that promote protective responses during pro-inflammatory conditions [14] and reduces the injury response and atherosclerotic development in arteries [15].

Oxidative stress is considered a crucial pathogenic factor for endothelial cell injury and death. The accumulation of reactive oxygen species, reactive nitrogen species, and cytokine activation is associated with many forms of vascular dysfunction [16]. One such marker is plasminogen activator inhibitor– 1 (PAI-1), which is upregulated in activated or injured endothelial [17] and vascular smooth muscle cells [18]. Increased expression of PAI-1 suppresses the fibrinolytic system and creates a prothrombotic state, resulting in the pathological deposition of fibrin and subsequent tissue damage. In vivo, this increased expression of PAI-1 by endothelial and smooth muscle cells is associated with the development of vascular thrombosis, fibrosis, and subsequent cardiovascular disease [17, 18].

It was previously demonstrated that COMP-4 –a novel nutraceutical compound consisting of ginger, L-citrulline, and the herbal components *Paullinia cupana* and muira puama– reverted the apoptosis, fibrosis, and oxidative stress observed in aging rats' cavernosal smooth muscle tissue, resulting in an improvement in the animals' erectile function [19]. It was further determined that COMP-4 induced these histological and physiological effects through activation and upregulation of NO and cGMP from the normally dormant iNOS-NO-cGMP pathway within the rat cavernosal smooth muscle cells [20]. COMP-4 was subsequently shown to have the same effect on the iNOS-NO-cGMP pathway in human cavernosal smooth muscle cells *in vitro* [21]. Moreover, COMP-4 has been shown to stimulate NO not only from iNOS induction but also by upregulating eNOS in osteoblasts, leading to accelerated fracture healing [22] and prevention of osteoporosis in ovariectomized rats [23].

Given the increasing evidence that COMP-4 stimulates the iNOS-NO-cGMP pathway in both rat and human cavernosal smooth muscle cells, along with its action on eNOS in osteoblasts, we sought to determine whether the compound would have a similar effect on the vascular endothelial cell and thereby affect endothelial function in general. To this end, we conducted an *in vitro* study to evaluate the impact of COMP-4 in a human arterial endothelial cell line with or without endothelial dysfunction, with a focus on cGMP expression, NO activity, NOS expression, and PAI-1 activity.

## Materials and methods

### Cell culture

Frozen human umbilical arterial endothelial cells (HUAEC, cat# 202-05n, Cell Applications, Inc., San Diego, CA) were thawed in a 37˚C water bath. The HUAEC used in our experiments were selected based on previously published work using DNA microarrays, which showed that these cells have similar characteristics to systemic arterial endothelial cells. Cells were

resuspended and dispensed into a coated T75 flask. Cells were seeded at a density of 5,000–7,000 cells/cm$^2$ in Human Meso Endo Growth Medium (cat# 212–500, Cell Applications), left undisturbed for 16 hours, and refreshed the following day with supplemented culture medium. The medium was changed every three days until the culture reached 90% confluency. Cells were then split and sub-cultured for conducting the experiments at a density of 3.8 x 10$^5$ cells per well at 100% confluency. To ensure consistent results, all experiments were conducted using cells in passages 2–5 to avoid senescence and alterations in cellular function.

## Reagents

COMP-4 is a mixture of four components created by combining muira puama (MP, 0.9 mg/ml), *Paullinia cupana* (PC, 0.9 mg/ml), ginger (G, 0.225 mg/ml), and L-citrulline (0.9 mg/ml). The MP, PC, and G were obtained from Naturex, South Hackensack, NJ. L-citrulline was obtained from Sigma-Aldrich, St. Louis, MO. The final concentration of the mixture is 0.69 mg/ml. PC, MP, and L-citrulline were prepared at a 100-fold stock solution, while ginger was prepared at a 200-fold stock solution due to solubility reasons. All components were dissolved in 70% ethanol, except for L-citrulline, which was dissolved in water. In the following experiments, 10 μl of the COMP-4 mixture was added per ml of media, representing an addition of 0.07% of ethanol. This concentration of ethanol is not considered lethal for the cells. The control cells received the same amount of ethanol (0.07%) as the treated cells in all the experiments [20].

For COMP-4, all four individual components were mixed, reaching a final concentration of 0.69 mg/ml. PC, MP, and L-citrulline were prepared at a 100-fold stock solution. Due to its solubility, G was prepared in a 200-fold stock solution, as previously described [20]. All the components were dissolved in 70% ethanol except for L-citrulline, which was dissolved in water. 10 μL of the COMP-4 mixture were added per mL of media. In all the experiments performed, control cells received the same amount of ethanol as treated cells [20].

In addition, several inhibitors of the NO-cGMP pathway were employed in the experiments. L-NIL, a specific inhibitor of iNOS (L-NIL hydrochloride, Cat# 80310, Cayman Chemical, Ann Arbor, MI), was used at 2 μM; L-NAME, a non-selective inhibitor of all three NOS isoforms (N$_\omega$-Nitro-L-arginine methyl ester hydrochloride, Cat# N575, Sigma Aldrich, St. Louis, MO), was used at 3 μM. IBMX, an inhibitor of the phosphodiesterase enzyme (3-Isobutyl-1-methylxanthine, Cat# I5879, Sigma Aldrich, St. Louis, MO), was used at 0.4 mM.

It has been shown that hydrogen peroxide ($H_2O_2$) is a crucial member of the class of reactive oxygen species (ROS), which is generated via the respiratory chain cascade and is a byproduct of cellular metabolism. $H_2O_2$ can also penetrate the plasma membrane and cause injury and apoptosis of endothelial cells [24], and as such, has been extensively used in *in vitro* models to induce endothelial dysfunction [25, 26]. Therefore, to generate a state resembling endothelial impairment, a series of experiments were conducted using $H_2O_2$ at a concentration of 100 μM (hydrogen peroxide solution 30% (w/w) in $H_2O$, containing stabilizer, Cat# H1009, Sigma Aldrich) [26]. The concentration of $H_2O_2$ was intentionally chosen to keep the effect sublethal to avoid cell death and observe functional impairment. All incubations with the vehicle, COMP-4, inhibitors, and $H_2O_2$ were conducted for 24 hours.

## Determination of cGMP expression

HUAEC were seeded at 3.8 x 10$^5$ cells per well in a 6-well plate, and after reaching confluency, cells were incubated with vehicle, COMP-4, IBMX, L-NIL, or L-NAME for 24 hours.

After incubation, the media were removed, and a 400 μL HCl 0.1 M was added for 20 minutes. Cells were then scraped, homogenized by pipetting, and centrifuged at 1,000g for 10

minutes. The supernatants were used to determine cGMP concentration by a colorimetric ELISA (cat# 581021, Cayman Chemical Company, Ann Arbor, MI), following the manufacturer's instructions. The enzymatic reaction product was determined by spectrophotometry at 405 nm absorbance and expressed as pmol/mg protein.

## Determination of nitrite formation

After incubation, the cell culture media were collected and frozen at -80°C prior to the determination of total nitrite concentration by the Griess reaction (cat# 780001, Cayman Chemical Company). Griess reagents R1 followed by R2 were added to the nitrite blanks, nitrite standards, and each sample on a 96-well plate. After 10 minutes of incubation at room temperature, the absorbance was measured with a plate reader at 540 nm, and the nitrite concentration was measured as previously described (Ferrini et al., 2019; Ferrini et al., 2021).

## Detection of intracellular nitric oxide formation by flow cytometry and by confocal microscopy

Intracellular NO production was determined by flow cytometry and confocal microscopy using the NO-specific probe 4-amino-5-methylamino-2',7'-difluorofluorescein diacetate (DAF-FM DA), a cell-permeable fluorescent probe for the detection of NO.

For confocal microscopy, HUAEC were seeded in 4-well chambers (Nunc™ Lab-Tek™, Thermo Fisher) at 40,000 cells per well and treated with or without COMP-4 for 4 and 24 hours. After incubation, the media was removed, and the cells were incubated with 5 $\mu$M DAF-FM DA (cat# D-23844, Thermo Fisher) in the dark for 40 min at 37°C. After washing with PBS 3 times, the counterstain DAPI was added to the chamber slides and incubated for 5 minutes. Green fluorescence staining from the DAF-FM DA in the cells was detected using a Leica SPE confocal microscope. Pictures were taken at 40X magnification, and the merged images of DAF-FM DA and DAPI were obtained with the Leica Application Suite X (LAS X) photo documentation system. These experiments were repeated twice.

For flow cytometry, HUAEC seeded at 1.0 x $10^6$ cells/ml in T25 flasks were treated with or without COMP-4 0.1M of the NO donor DETA NONOate (cat# 82120, Cayman Chemical), and 5μg/mL lipopolysaccharide (LPS, cat # 00-4976-93, Invitrogen eBioscience.) for 24 hours. Cells were then incubated for 2 hours with 5 μM DAF-FM DA. Following incubation and washing, cells were treated with trypsin-EDTA to detach gently before immediate flow cytometry analysis. During this part of the experiment, exposure to light was avoided due to the light sensitivity of the probe. The intracellular fluorescence of DAF-FM DA was analyzed by flow cytometry using Attune NxT (Thermo Fisher; excitation wavelength 488 nm, emission wavelength 519 nm). The results were expressed as percent DAF-FM DA in live cells.

## Intracellular nitric oxide activity

To corroborate the results obtained by flow cytometry, another system for detecting intracellular NO was used, the GUAVA Muse$^®$ Nitric Oxide Kit (cat# MCH100112, Luminex, Austin, TX). HUAEC were seeded at 1.0 x $10^6$ cells/ml in T25 flasks, treated with vehicle, COMP-4 with or without L-NAME, or LPS (positive control), and incubated for 24 hours. Cells were then suspended in an assay buffer at $10^6$ cells/ml. 100 μL of a working solution consisting of the Muse Nitric Oxide Reagent (DAX-J2 Orange) was added to 10 μL of cells in suspension. Samples were incubated for 30 minutes in a 37°C incubator with 5% $CO_2$. After the incubation, 90 μL of the Muse 7-AAD working solution (dead cell marker) was added to each tube and incubated at room temperature for 5 min, protected from light. Intracellular NO was measured using the Guava® Muse Cell Analyzer, calibrated with negative and positive controls.

Generated plots were divided into quadrants of nitric oxide positivity (+ or -) and dead cells (+ or -). From each plot, the lower-right quadrant of nitric oxide (+) and dead cell marker (-) was used to express intracellular NO results as the percent positive NO in live cells, which was compared between different treatments.

## Western blotting and densitometry analysis

After incubating with vehicle and treatments, 30 μg of protein obtained from the cell lysates were subjected to gel electrophoresis with 4–15% Tris–HCl PAGE (Bio-Rad, Hercules, CA) in running buffer (Tris/glycine/SDS). The proteins were transferred onto polyvinylidene fluoride (PVDF) membranes embedded in transfer buffer (Tris/glycine/methanol) using transblot semi-dry apparatus (Bio-Rad, Hercules CA). The nonspecific binding was blocked by immersing the membranes into 5% nonfat dried milk and 0.1% (v/v) Tween 20 in PBS for 1 hour at room temperature, as previously described (3,18). After several washes with the washing buffer (PBS Tween 0.1%), the membranes were incubated with the primary antibodies overnight at 4˚C. Primary mouse monoclonal and rabbit polyclonal antibodies were used for: iNOS at 1:250 dilution (Abcam cat# ab15323, RRID:AB_301857, Cambridge, UK), eNOS at 1:500 dilution (BD Biosciences cat# 610299, RRID:AB_397693, San Jose, CA); nNOS at 1:500 dilution (Abcam cat# ab76067, RRID:AB_2152469), GAPDH at 1:5000 dilution (Millipore Sigma cat# MAB374, RRID:AB_2107445, Ontario, Canada), and PAI-1 at 1:2000 dilution (Abcam cat# ab182973). The membranes were incubated for 2 hours at room temperature with 1:2000 dilution of anti-mouse or anti-rabbit secondary antibody linked with HRP (Cell Signaling Technology, cat# 7076, RRID:AB_330924 and cat# 7074, RRID:AB_2099233, Danvers, MA). After several washes, the bands were visualized using the WesternSure PREMIUM chemiluminescent detection system (Li-COR Biotechnology cat# 926–95000, Lincoln, NE). The bands of the respective analyte and the total protein content in each well were scanned by LI-COR Odyssey Fc Infrared Imaging System (LI-COR) and semi-quantified using ImageStudio (v5.2, LI-COR). The results were expressed as analyte/total protein normalized with respect to the control.

## Quantitative real-time RT-PCR to evaluate expression of NOS isotypes

RNA was extracted using Trizol Reagent (Invitrogen, Carlsbad, CA), and equal amounts (1 μg) of RNA were reverse transcribed using a high-capacity RNA-to-cDNA PCR kit (Applied Biosystems, Foster City, CA). The PCR primer set (RT$^2$) for human inducible nitric oxide synthase (iNOS), human neuronal nitric oxide synthase (nNOS), and human endothelial nitric oxide synthase (eNOS) were obtained from SABiosciences (Germantown, MD). Real-time PCR (SYBR GreenER, Applied Biosystems) was performed using a StepOne Plus instrument (Applied Biosystems). The protocol included melting for 15 min at 95˚C, 40 cycles of three-step PCR including melting for 15 s at 95˚C, annealing for 30 s at 58˚C, elongation for 30 s at 72˚C with an additional detection step of 15 s at 81˚C, followed by a melting curve from 55 to 95˚C at the rate of 0.5˚C per 10 s. Experimental mRNA starting quantities were then calculated from the standard curves and averaged using SABiosciences software as previously described [21, 27]. The ratios of the experimental marker gene (e.g., iNOS, eNOS, or nNOS mRNA) to housekeeping gene RPLP1/3 mRNA were computed and normalized with control samples.

## Cytokine array expression

A cytokine/inflammation antibody array (R&D Systems, Minneapolis, MN) was used to assess the modulation of cytokines in HUAEC by COMP-4 and $H_2O_2$. The array was performed by following the manufacturer's instructions. Briefly, HUAEC were cultured on 6-well plates and

treated for 24 hours with vehicle, COMP-4, the NO donor DETA NONOate, LPS (positive control of cytokine stimulation), $H_2O_2$, or the combination $H_2O_2$ + COMP-4, prior to overnight serum starvation. The supernatant (500 µL) was collected and incubated overnight with pre-blocked membranes spotted in duplicates with 36 antibodies for a variety of cytokines, chemokines, and acute-phase protein. An IR Dye 800CW Streptavidin (LI-COR, Catalog #926–32230) at 1:2000 dilution in array buffer was added to the membranes for 30 minutes at room temperature on a rocking platform. After several washes, the excess buffer was removed from the membranes by blotting the lower edge onto absorbent paper. The blotting images were scanned by LI-COR Odyssey FC Infrared Imaging System (LI-COR, Lincoln, NE), and cytokines were semi-quantified using ImageStudio (v5.2, LI-COR). Quantification was done using the LI-COR photo-documentation system, which uses total protein normalization instead of loading control. It has been shown that total protein normalization is more accurate than housekeeping protein expression, the latter of which is more variable than constant and changes with cell type and developmental stage.

## PAI-1 activity

To assess the influence of COMP-4 on PAI-1 activity, HUAEC were seeded at equal densities in each well and grown on 6-well cell culture plates until they reached confluence. The monolayer of cells was incubated with vehicle or COMP-4 with or without $H_2O_2$ for 24 hours. The cell culture medium was collected and centrifuged at 3,000g for 15 min and 4°C to remove debris. The PAI-1 Activity Assay Kit from AbCam (cat# 283368) was employed following the manufacturer's instructions. In brief, this is a two-step colorimetric assay for PAI-1. Samples are first incubated with a known amount of tissue plasminogen activator (tPA), allowing PAI-1 and tPA to form an inactive complex. Residual free tPA in the sample is exposed to a reaction mixture containing plasminogen and a chromogenic substrate cleaved by plasmin. Plasminogen is converted by the free tPA in the sample to plasmin, which acts on the substrate to release p-nitroaniline (p-NA). The absorbance of the released p-NA is inversely proportional to PAI-1 activity in the samples. Samples were assayed in duplicate.

## Statistical analysis

All data are presented as mean ± S.E.M. Differences between groups were analyzed by one-way ANOVA followed by Tukey's multiple comparisons tests or between two groups by unpaired Student's t-test using GraphPad Prism version 9.0.0 (GraphPad Software, San Diego, CA). All comparisons were two-tailed, and $p < 0.05$ was considered statistically significant. Using simple randomization, cells were seeded in 6-well plates at 3.8 x $10^5$ cells per well or 4-well chambers at 40,000 cells per well. All in vitro experiments were repeated three times, and data from representative experiments are shown.

## Results

### COMP-4 upregulates cGMP and NO production in HUAEC

To determine whether the NO-cGMP pathway is functional in this cell line and whether COMP-4 has a regulatory effect on it, the expression of cGMP was studied. **Fig 1A** shows that there was endogenous production of cGMP in this cell line, along with an increase in cGMP expression after treatment with the phosphodiesterase inhibitor IBMX. When the cell line was incubated with the non-selective NOS inhibitor L-NAME, cGMP expression was reduced by 40%, whereas when incubated with the selective iNOS inhibitor L-NIL, there was no inhibition of cGMP expression. **Fig 1B** shows that incubation of the HUAEC with COMP-4 alone

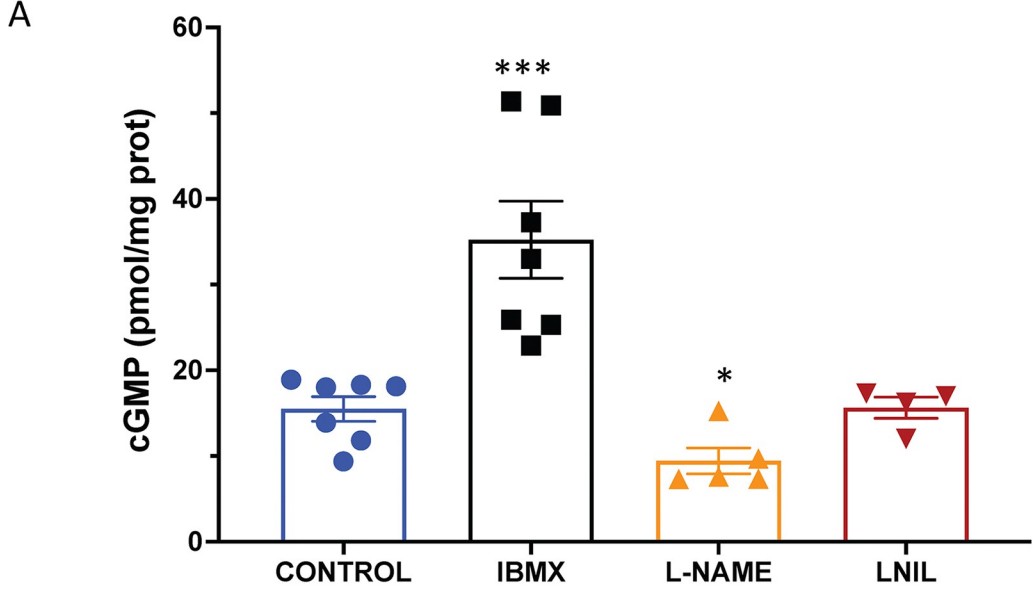

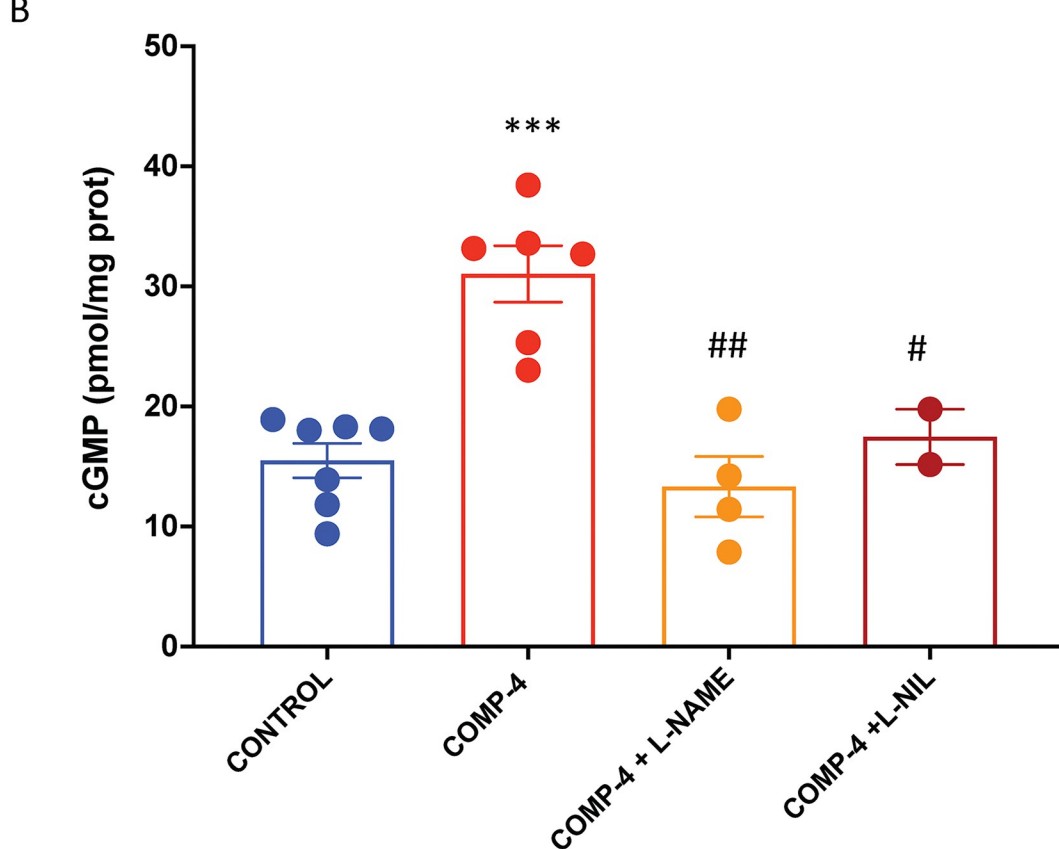

**Fig 1. Effect of the incubation of COMP-4 on cGMP expression in human umbilical arterial endothelial cells (HUAEC).** A) HUAEC cells were incubated for 24 hours with vehicle (CONTROL), IBMX as a positive control for downstream cGMP expression, L-NAME as a non-selective inhibitor of the three nitric oxide synthases, and L- NIL as a selective inhibitor of iNOS. Afterward, cGMP was measured by colorimetric ELISA. Results for cGMP quantitation are expressed as pmol/mg protein and represent the mean ± S.E.M. of two and four experiments done in duplicates. Statistical differences were examined by one-way

ANOVA followed by Tukey multiple comparison test. *p<0.05, ***p<0.001 vs. control. B) HUAEC cells were incubated for 24 hours with or without COMP-4 (0.69 mg/ml), as well as co-incubation of COMP-4 with L-NAME and L-NIL. Results for cGMP quantitation are expressed as pmol/mg protein and represent the mean ± S.E.M. of two and four experiments done in duplicates. Statistical differences were examined by one-way ANOVA followed by Tukey multiple comparison test. **p<0.01; ***p<0.001 vs. control; #p<0.05; ##p<0.01 with respect to COMP-4.

increased cGMP expression by 2-fold with respect to control, while co-incubation of COMP-4 with either L-NAME or L-NIL significantly reduced the cGMP expression by 57% and 44%, respectively, compared to COMP-4 incubation alone.

The production of NO by these same incubations was measured a) by nitrite formation in the cell culture media, b) intracellularly by DAF DA staining and flow cytometry, and c) by the use of the Luminex MUSE Nitric Oxide Kit. **Fig 2A** shows that COMP-4 markedly increased nitrite formation in cell culture media compared to control (p<0.001). Because these absolute nitrite levels in the media of these human endothelial cells were much lower than those observed previously in human cavernosal smooth muscle cells, we also studied the expression of NO intracellularly by three different methods. **Fig 2B** shows the expression of DAF-2T staining, the product of the oxidation of DAF DA with NO, in the HUAEC cells with or without COMP-4 treatment at two time points. Positive staining of DAF-2T (green) was observed with COMP-4 treatment at 4 and 24 hours, whereas untreated controls showed faint expression of DAF-2T, and only the DAPI counterstain (blue) was observed in the merged control confocal images. The expression of DAF-2T was also studied by flow cytometry. Using LPS and NO donor DETA NONOate as positive controls for this experiment, **Fig 2C** shows that the percent of DAF-2T positive live cells incubated with COMP-4 was upregulated with respect to untreated controls, and expression was similar to the level achieved by treatment with LPS and DETA NONOate. The same effect of COMP-4 on intracellular NO production was observed using another intracellular NO detection probe, DAX-J2 Orange (**Fig 2D**). Not unexpectedly, the addition of L-NAME to COMP-4 significantly reduced the expression of DAX-J2 Orange live cells by 46% when compared to COMP-4 alone. Since the primary objective was to examine intracellular NO production in response to COMP-4 treatment, independent of downstream effects on cGMP degradation, IBMX was not included in the experiments described in **Fig 2**.

## COMP-4 increases eNOS and iNOS expression in HUAEC cells

To determine whether the increase in NO production and cGMP expression observed with COMP-4 within the HUAEC could be due to changes in the expression of any of the three endogenous nitric oxide synthases, real-time RT-PCR and western blots were performed on these cells treated with or without COMP-4. **Fig 3A** shows that COMP-4 significantly increased eNOS mRNA expression by 4.4-fold (p<0.01) and iNOS mRNA expression by 3.9-fold (p = 0.0102) with respect to untreated controls. The expression of nNOS was not changed by COMP-4, and the observed in nNOS was not statistically significant. The protein concentrations for the three NOS enzymes were also studied by western blot. **Fig 3B** shows that treatment with COMP-4 increased the expression of eNOS by 5-fold and iNOS by 2.5-fold; nNOS expression was unchanged.

## COMP-4 reduces cytokine expression in HUAEC

To determine whether COMP-4, by increasing NO production, can act as an anti-inflammatory/antifibrotic compound by modulating the expression of cytokines, the level of various cytokines released in the cell culture media in the presence or absence of COMP-4 was studied.

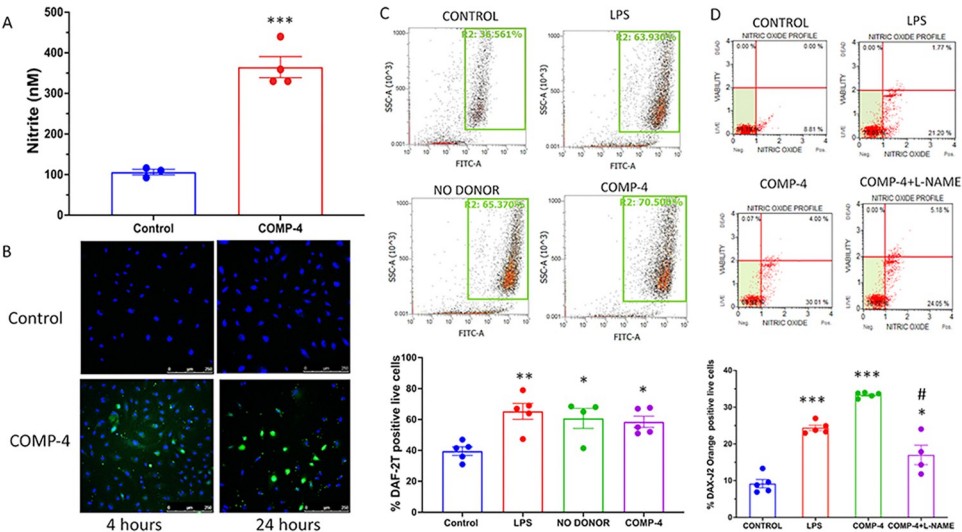

**Fig 2. Effect of the incubation of COMP-4 on nitric oxide formation in HUAEC.** A) Nitrite production in the cell culture media. HUAEC were incubated for 24 hours with or without COMP-4. The cell culture media were collected and frozen at -80C. Nitrite formation was determined by the Griess reaction. Results are expressed as nM and represent the mean ± S.E.M. of four experiments done in duplicates. Statistical differences were examined by unpaired Student's t-test. ***$p<0.001$. B) Intracellular NO detection by confocal microscopy using a DAF-FM probe. HUAEC were incubated with or without COMP-4 for 4 hours and 24 hours. The green fluorescence signal represents DAF-FM DA. DAPI counterstain. Magnification 200X. C) Intracellular detection of NO by flow cytometry using a DAF-FM probe. HUAEC cells were incubated with COMP-4 for 24 hours. LPS and the NO donor, DETA NONOate, were used as positive controls of NO formation. Results are expressed as % of DAF-2 positive live cells and represent the mean ± S.E.M of two experiments done in duplicates. Statistical differences were examined by one-way ANOVA followed by Tukey multiple comparison test. *$p<0.05$; **$p<0.01$ vs. control. D) Intracellular detection of NO by the Muse® Nitric Oxide Kit. HUAEC cells were incubated with COMP-4. The kit uses DAX-J2 Orange reagent to detect NO formation along with a marker of cell death, 7-AAD. HUAEC were incubated with or without COMP-4. LPS was used as a positive control of NO formation. L-NAME was used as a non-selective inhibitor of NOS and co-incubated with COMP-4 for 24 hours. Results are expressed as % of DAX-J2 positive live cells and represent the mean ± S.E.M of two experiments done in duplicates. Statistics: one-way ANOVA followed by Tukey multiple comparison test. **$p<0.01$; ***$p<0.001$ vs. control. #$p<0.05$ with respect to COMP-4.

**Fig 4** shows that after treatment of HUAEC with COMP-4 for 24 hours, there was a decrease in the expression of PAI-1 and IL-8 compared to untreated controls. The NO donor, DETA NONOate, used as a positive control for NO release, showed a similar expression profile as that for COMP-4.

## COMP-4 prevents impairment in endothelial function induced by $H_2O_2$

Since hydrogen peroxide ($H_2O_2$) is considered the primary source of endogenous ROS and has been extensively used to induce endothelial dysfunction *in vitro*, we further investigated whether COMP-4, by increasing NO production, can improve such $H_2O_2$-induced endothelial dysfunction in HUAEC. **Fig 5** shows that $H_2O_2$ decreases nitrite formation while co-incubation of $H_2O_2$ with COMP-4 increases nitrite formation by 3-fold with respect to $H_2O_2$ alone.

In addition, to further test the hypothesis that COMP-4 can attenuate endothelial dysfunction caused by $H_2O_2$, the cytokine profiles in cell lysates were studied. **Fig 6** shows that $H_2O_2$ increased the expression of IL-6, IL-8, MIF, PAI-1, and CXCL-1/GRO, while the co-incubation of $H_2O_2$ with COMP-4 decreased cytokine expression, similar to the levels achieved with COMP-4 alone.

We further investigated the expression of PAI-1 due to its critical role in atherothrombotic diseases, coronary artery disease, and myocardial infarction. The reduction of the expression

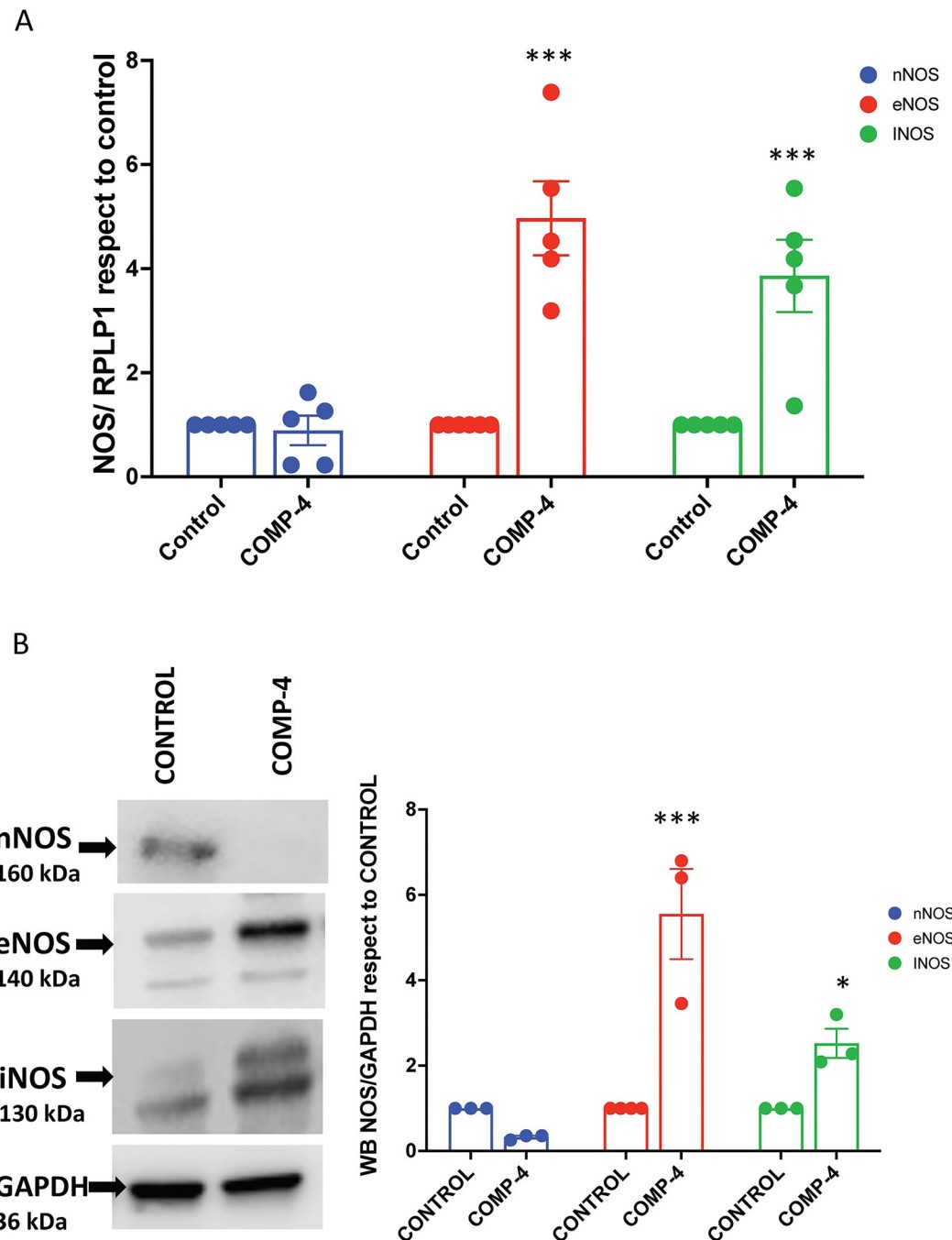

**Fig 3. Effect of COMP-4 on mRNA and protein expression of nNOS, eNOS, and iNOS.** HUAEC cells were incubated with COMP-4 for 24 hours. A) mRNA expression for nNOS, eNOS, and iNOS was determined by qPCR. Results were expressed as a fold increase with respect to the control. ***$p<0.001$ with respect to eNOS and iNOS control. B) Protein expression for nNOS, eNOS, and iNOS was determined by western blot. Results were expressed as ratio NOS/GAPDH with respect to the control for each isoenzyme. Statistical differences were examined by unpaired Student's t-test. *$p<0.05$ and ***$p<0.001$ with respect to control for eNOS and iNOS.

of PAI-1 was corroborated by western blot in cell lysates and the media. **Fig 7A** shows that COMP-4 treatment reduced the expression of PAI-1 in the cell lysate by 32% (p = 0.0063) and in the media by 32% p = 0.0292). Moreover, the expression of PAI-1 was upregulated by $H_2O_2$

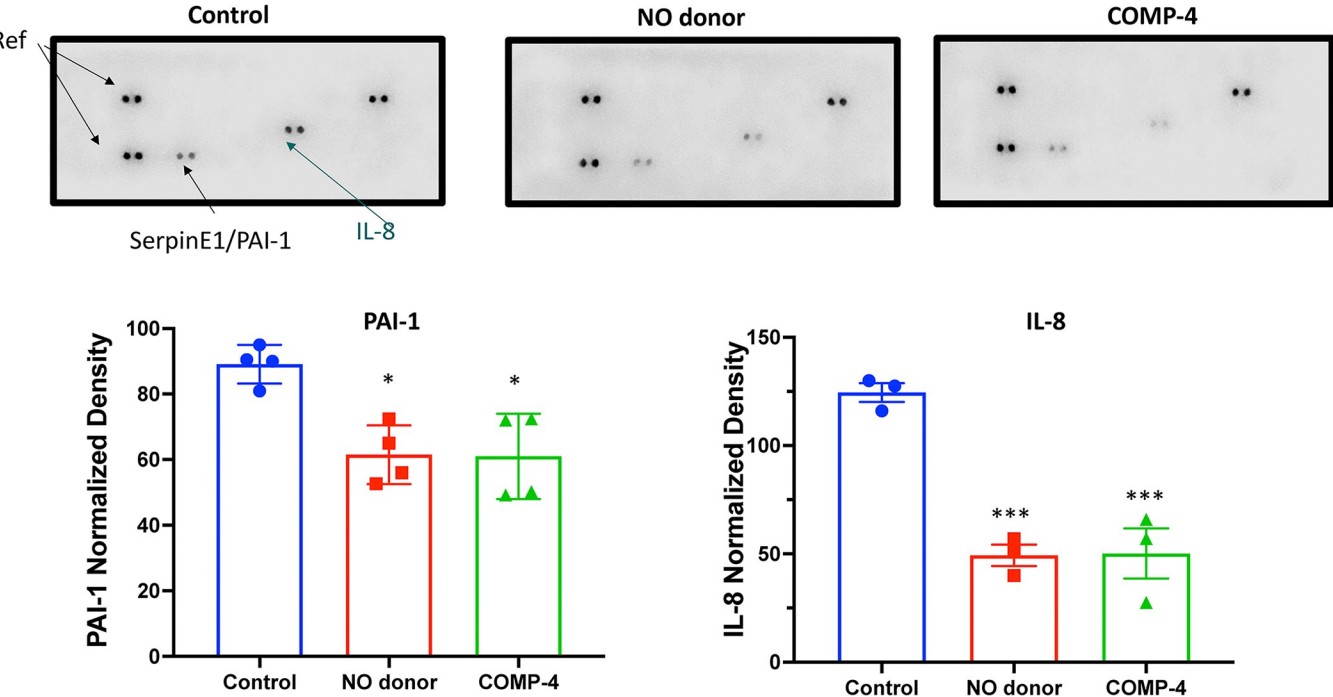

**Fig 4. Effect of COMP-4 on pro-inflammatory cytokine expression in HUAEC.** Cytokine array analysis was performed using the cell culture media of HUAEC cells treated with or without COMP-4 for 24 hours. Only two cytokines out of forty in the array (IL-8 and PAI-1) were expressed in the control group. DETA NONOate was used as NO donor. Results were expressed as normalized density regarding the reference spots and represent the mean ± S.E.M. of two duplicated experiments. Statistical differences were examined by one-way ANOVA followed by Tukey multiple comparison test. *p<0.05 and ***p<0.001 with respect to control.

and down-regulated by the co-incubation of COMP-4 with $H_2O_2$. The same results were observed by measuring the secreted PAI-1 activity in **Fig 7B**. A reduction of PAI-1 activity by 42% (p = 0.092) was observed after the co-incubation of $H_2O_2$ with COMP-4.

## Discussion

The present study demonstrates that in human arterial endothelial cells, the compound COMP-4 is capable of activating the eNOS/iNOS-NO-cGMP pathway, leading to an increase in the production and bioavailability of NO, as well as protection of endothelial cells from $H_2O_2$-induced injury.

Increases in NO and cGMP in human endothelial cells appear to be due to COMP-4's activation of eNOS and iNOS but not nNOS as evidenced by the increase in both the mRNA and protein content of the eNOS and iNOS enzymes. This was further confirmed by the observation that L-NAME, a non-selective NOS inhibitor, as well as L-NIL, a specific inhibitor of iNOS, are both capable of blocking cGMP formation by COMP-4.

Previous work has shown that NO has an anti-inflammatory effect through the downregulation of pro-inflammatory cytokines [28]. In the present study, COMP-4 treatment was shown to abrogate $H_2O_2$-induced expression of MIF, IL-6, IL-8, and CXCL1, suggesting that COMP-4 can reduce inflammation, likely by increasing NO availability. Further work is needed to elucidate the mechanism by which COMP-4 acts to reduce cytokine expression since the release of pro-inflammatory cytokines is regulated in part by transcription factors such as nuclear factor-κB (NF-κB) [29]. Whether COMP-4 acts directly or indirectly through NO release in activating NF-κB or through other mediators require further study.

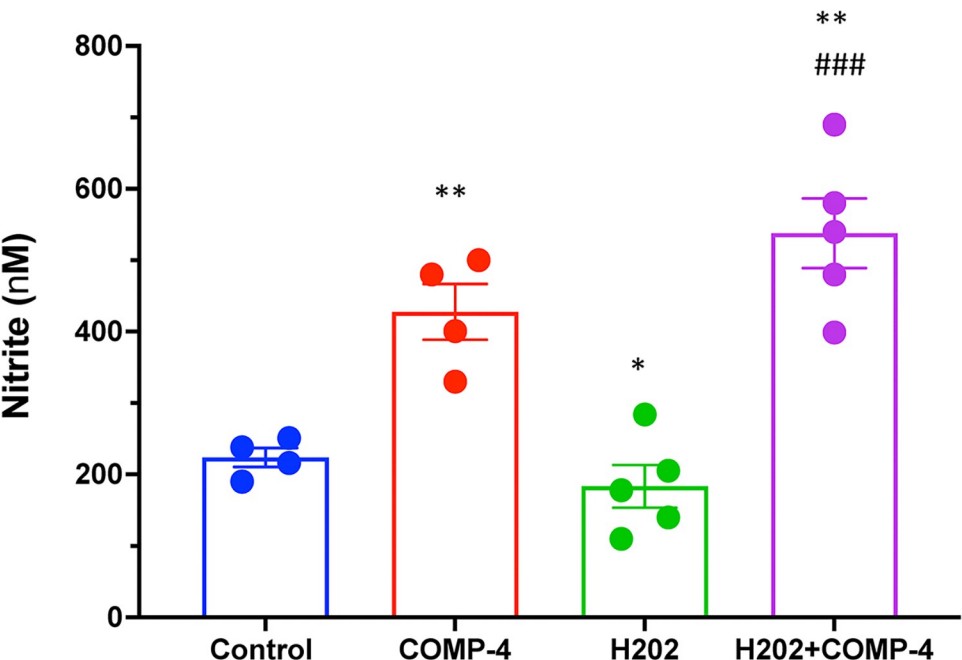

**Fig 5. Effect of COMP-4 in mitigating H2O2-induced endothelial damage.** Cells were treated with or without COMP-4 plus 100 μM H2O2 for 24 hours. Co-incubation with COMP-4 and H2O2 increased nitrite formation. Statistical differences were examined by one-way ANOVA followed by Tukey multiple comparison test. *p<0.05; **p<0.01 with respect to control. ###p<0.001 with respect to H2O2.

COMP-4 treatment also appeared to attenuate $H_2O_2$-induced impairment in endothelial function by modulating nitrite formation, a footprint of NO production. Note that the concentration of $H_2O_2$ (100 μM) was intentionally chosen to keep the effect sublethal to avoid cell death while still expecting functional impairment. The observed results suggest that COMP-4's effect on eNOS and iNOS expression may play a cytoprotective role by increasing NO production and bioavailability.

As further evidence of a protective role for COMP-4, treatment of cells with COMP-4 reduced the expression and activity of PAI-1 in the setting of $H_2O_2$-induced endothelial injury. PAI-1 is mainly produced by the endothelium and has a multifaceted role in inflammation [30, 31], oxidative stress [32], fibrosis [33], and macrophage adhesion/migration [34, 35]. The PAI-1 results from this study suggest that COMP-4 may modulate tissue fibrosis and remodeling even after endothelial dysfunction has already occurred. This result is consistent with our prior observation in rat cavernosal smooth muscle with long-term treatment of senescent tissue with COMP-4 [19].

This study is not without limitations. For example, we have not identified whether the endothelial effects observed with COMP-4 are due to an individual component of the compound, or the synergistic effect of all four components. However, previous results in cavernosal smooth muscle cells showed that the complete COMP-4 compound rather than any of the individual four components had the most profound effect in modulating each one of the specific steps within the iNOS-NO-cGMP pathway [20, 21], as well as in reducing oxidative stress markers in the penile corpora cavernosa [36]. Whether the results from smooth muscle cells are translatable to the endothelial cell model requires further investigation.

Taken together, the present study shows that the nutraceutical compound COMP-4, a product consisting of ginger, L-citrulline, muira puama, and *Paullinia cupana*, may have a

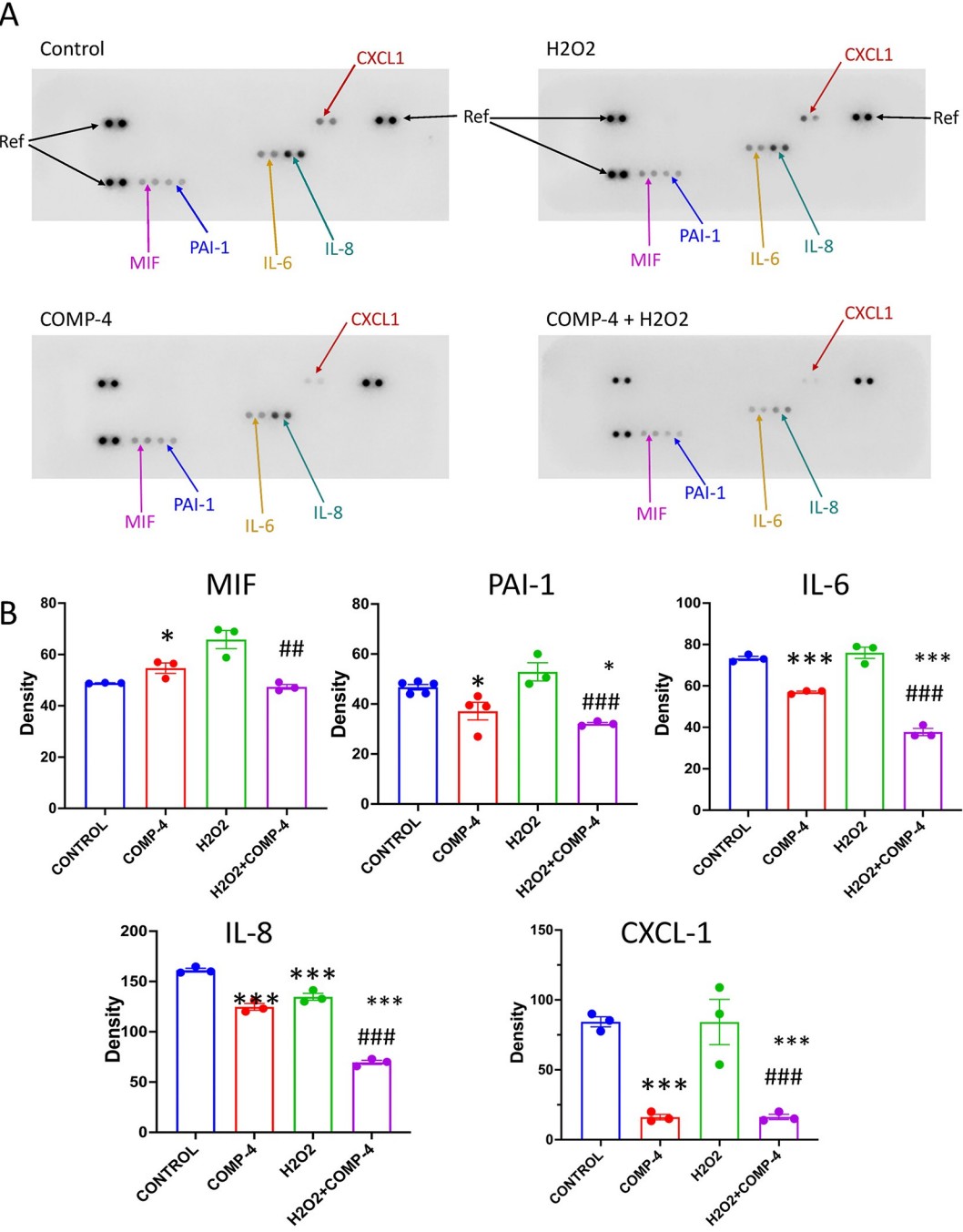

**Fig 6. Effect of COMP-4 on the expression of cytokines in H2O2-treated cells.** Cytokine array analysis was performed using the cell culture media of HUAEC cells treated with or without COMP-4 and with the addition of H2O2 for 24 hours. Five out of forty cytokines in the array (MIF, IL-8, IL-6, CXCL1, and PAI-1) were expressed in the H2O2 group, and co-incubation with COMP-4 reduced the expression of these cytokines. Results are expressed as normalized density regarding the reference spots and represent the mean ± S.E.M. of three duplicated experiments. Statistical differences were examined by one-way ANOVA followed by Tukey multiple comparison test. *p<0.05; **p<0.01; ***p<0.001 with respect to control. ## p<0.01 ### p<0.001 with respect to H2O2.

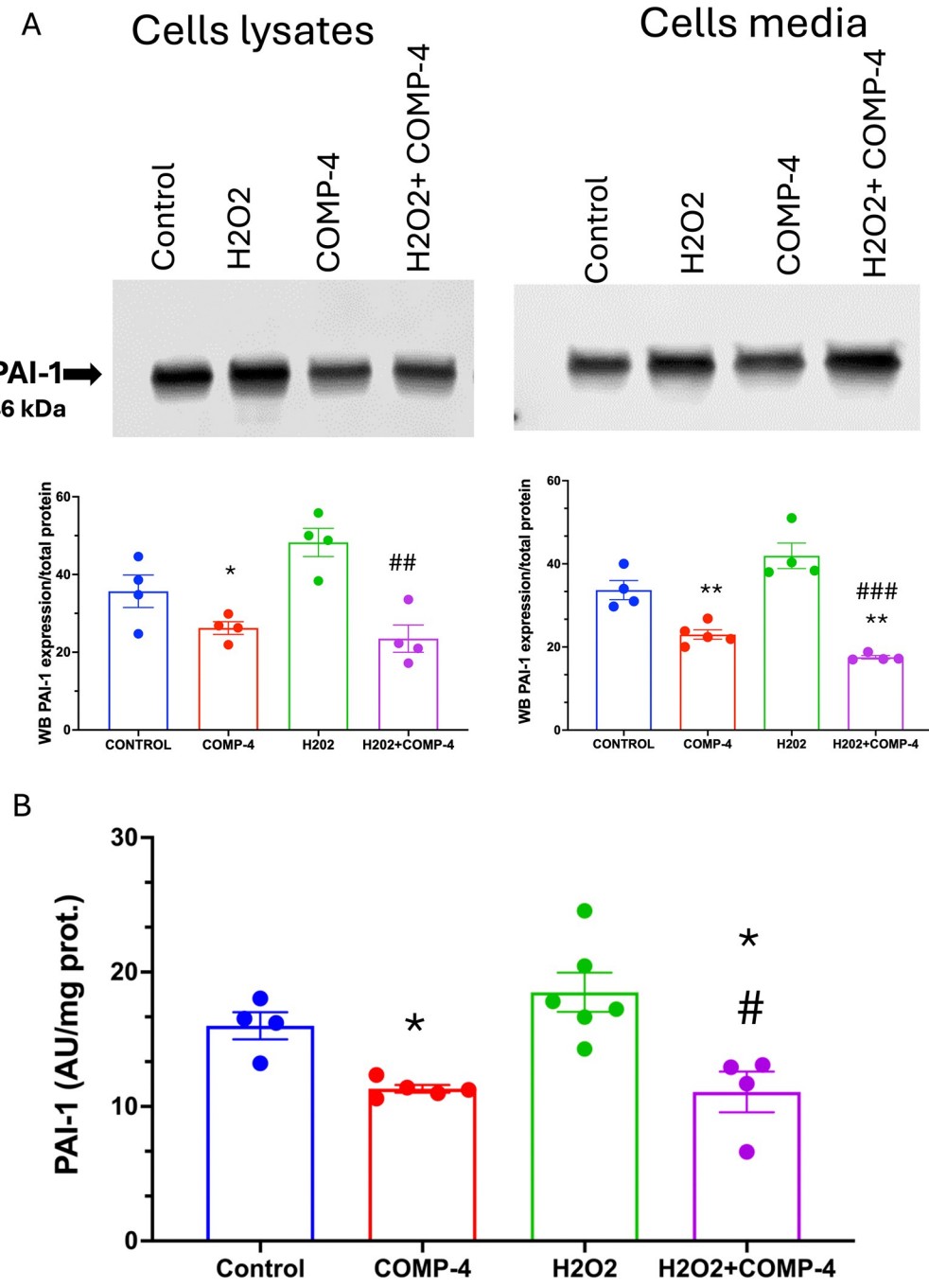

**Fig 7. Effect of COMP-4 on PAI-1 expression and activity in $H_2O_2$-treated cells.** A) PAI-1 expression was determined by western blot in cell homogenate, and the supernatant of the HUAEC treated with H2O2 with or without COMP-4 for 24 hours. Results are expressed as normalized density with respect to total protein per lane and represent the mean ± S.E.M. of two duplicated experiments. Statistical differences were examined by one-way ANOVA followed by Tukey multiple comparison test. *p<0.05; **p<0.01 with respect to control. ## p<0.01 ### p<0.001 with respect to H2O2. B) PAI-1 activity was quantified using an indirect colorimetric assay. H2O2-treated HUAEC showed increased PAI-1 activity, which was abrogated by COMP-4 treatment. Results are expressed as activity unit (AU) per mg protein and represent the mean± S.E.M. of three experiments done in duplicate. Statistical differences were examined by one-way ANOVA followed by Tukey multiple comparison test. *p<0.05 with respect to control, #p<0.05 versus H2O2 treatment alone.

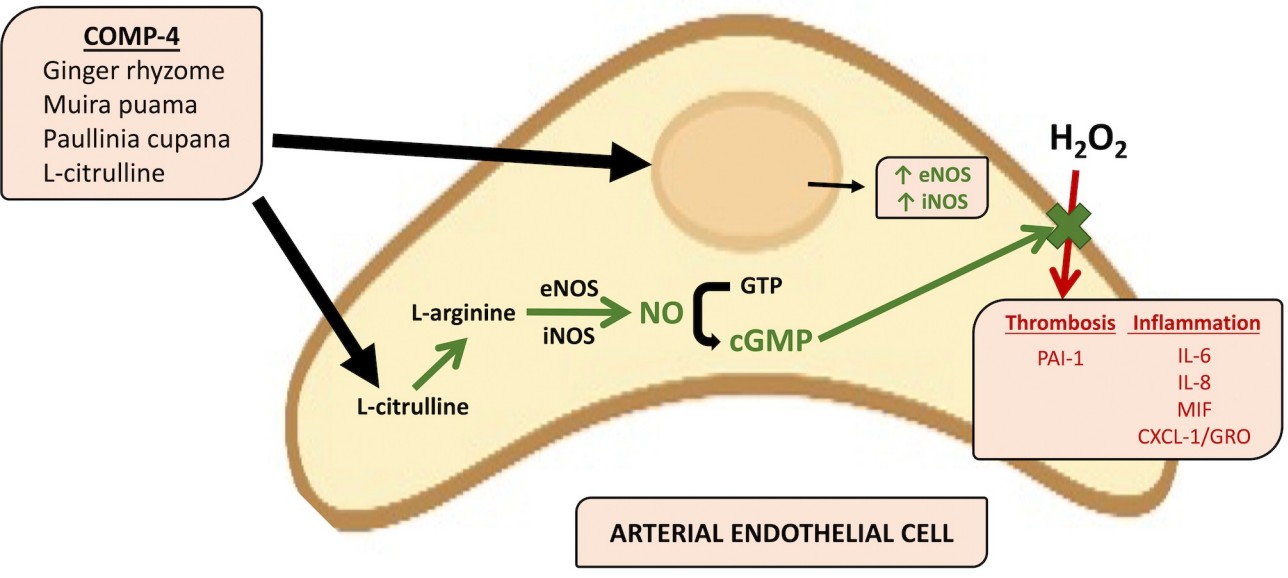

**Fig 8. Proposed pathway for the effects of COMP-4 in the human arterial endothelial cell.**

protective role against arterial endothelial dysfunction. This effect is accomplished through activation of the eNOS/iNOS-NO-cGMP pathway and ameliorates the pro-inflammatory milieu induced by $H_2O_2$ injury. **Fig 8** summarizes a proposed pathway for the activity of COMP-4 on the human arterial endothelial cell. Furthermore, the effect of COMP-4 treatment on PAI-1 expression and activity in response to cell injury suggests that there may be a protective role for COMP-4 or other nutraceutical or herbal compounds that target NO production. COMP-4 has been studied in humans and has an innocuous side effect profile [37]. A human subject trial to confirm efficacy in combating endothelial dysfunction appears warranted.

## Supporting information

**S1 Fig. Western blot data for eNOS expression.**
(PDF)

**S2 Fig. Western blot data for iNOS expression.**
(PDF)

**S3 Fig. Western blot data for nNOS expression.**
(PDF)

**S4 Fig. Gels and quantification data for PAI-1 results in Fig 7.**
(PDF)

**S1 Table. cGMP expression data comprising Fig 1 results.**
(PDF)

**S2 Table. Data points comprising Fig 2A.**
(PDF)

**S3 Table. Data points comprising Fig 2B.**
(PDF)

**S4 Table. Data points comprising Fig 2C.**
(PDF)

**S5 Table. Data points comprising Fig 2D.**
(PDF)

**S6 Table. Data points comprising Fig 3A.**
(PDF)

**S7 Table. Data points comprising Fig 3B.**
(PDF)

**S8 Table. Data points comprising Fig 4.**
(PDF)

**S9 Table. Data points comprising Fig 5.**
(PDF)

**S10 Table. Data points obtained from analysis of gels in Fig 6A.**
(PDF)

**S11 Table. Data points obtained from analysis of gels in Fig 6A.**
(PDF)

**S12 Table. Data points obtained from analysis of gels in Fig 6A.**
(PDF)

**S13 Table. Data points obtained from analysis of gels in Fig 6A.**
(PDF)

**S14 Table. Data points obtained from analysis of gels in Fig 6A.**
(PDF)

**S15 Table. Data points obtained from analysis of gels in Fig 7.**
(PDF)

**S16 Table. Data points obtained from analysis of gels in Fig 7.**
(PDF)

**S17 Table. Data points obtained from analysis of gels in Fig 7.**
(PDF)

## Author Contributions

**Conceptualization:** Monica G. Ferrini, Jacob Rajfer.

**Data curation:** Monica G. Ferrini, Andrea Abraham, Revecca Millán, Leslie Graciano, Jacob Rajfer.

**Formal analysis:** Monica G. Ferrini, Sriram V. Eleswarapu.

**Investigation:** Monica G. Ferrini.

**Methodology:** Monica G. Ferrini, Andrea Abraham, Revecca Millán, Leslie Graciano.

**Project administration:** Monica G. Ferrini, Sriram V. Eleswarapu, Jacob Rajfer.

**Software:** Monica G. Ferrini.

**Supervision:** Monica G. Ferrini, Sriram V. Eleswarapu, Jacob Rajfer.

**Writing – original draft:** Monica G. Ferrini.

**Writing – review & editing:** Sriram V. Eleswarapu.

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
