## [Decision Letter · Decision Letter 0]

10 Sep 2024

PONE-D-24-27761Nutraceutical COMP-4 confers protection against endothelial dysfunction through the eNOS/iNOS-NO-cGMP pathwayPLOS ONE

Dear Dr. Eleswarapu,

Thank you for submitting your manuscript to PLOS ONE. After careful consideration, we feel that it has merit but does not fully meet PLOS ONE’s publication criteria as it currently stands. Therefore, we invite you to submit a revised version of the manuscript that addresses the points raised during the review process.

We look forward to receiving your revised manuscript.

Kind regards,

Yung-Hsiang Chen, Ph.D.

Academic Editor

PLOS ONE

Journal Requirements: When submitting your revision, we need you to address these additional requirements. 1. Please ensure that your manuscript meets PLOS ONE's style requirements, including those for file naming. The PLOS ONE style templates can be found at https://journals.plos.org/plosone/s/file?id=wjVg/PLOSOne_formatting_sample_main_body.pdf and https://journals.plos.org/plosone/s/file?id=ba62/PLOSOne_formatting_sample_title_authors_affiliations.pdf 2. We note that the grant information you provided in the ‘Funding Information’ and ‘Financial Disclosure’ sections do not match.  When you resubmit, please ensure that you provide the correct grant numbers for the awards you received for your study in the ‘Funding Information’ section. 3. Thank you for stating the following financial disclosure: "This work was supported by a grant from the Peter A. Morton Foundation; and the NIH National Institute on Minority Health and Disparities (NIH/NIMHD) [grant number U54MD007598 to MGF]." Please state what role the funders took in the study.  If the funders had no role, please state: ""The funders had no role in study design, data collection and analysis, decision to publish, or preparation of the manuscript."" If this statement is not correct you must amend it as needed. Please include this amended Role of Funder statement in your cover letter; we will change the online submission form on your behalf. 4. Thank you for stating the following in the Competing Interests section: "I have read the journal's policy and the authors of this manuscript have the following competing interests: JR is a principal in KLRM, LLC, which holds the patent for COMP-4. All other authors have declared that no competing interests exist."  Please confirm that this does not alter your adherence to all PLOS ONE policies on sharing data and materials, by including the following statement: ""This does not alter our adherence to  PLOS ONE policies on sharing data and materials.” (as detailed online in our guide for authors http://journals.plos.org/plosone/s/competing-interests).  If there are restrictions on sharing of data and/or materials, please state these. Please note that we cannot proceed with consideration of your article until this information has been declared.  Please include your updated Competing Interests statement in your cover letter; we will change the online submission form on your behalf. 5. We note that your Data Availability Statement is currently as follows: All relevant data are within the manuscript and its Supporting Information files. Please confirm at this time whether or not your submission contains all raw data required to replicate the results of your study. Authors must share the “minimal data set” for their submission. PLOS defines the minimal data set to consist of the data required to replicate all study findings reported in the article, as well as related metadata and methods (https://journals.plos.org/plosone/s/data-availability#loc-minimal-data-set-definition). For example, authors should submit the following data: - The values behind the means, standard deviations and other measures reported;- The values used to build graphs;- The points extracted from images for analysis. Authors do not need to submit their entire data set if only a portion of the data was used in the reported study. If your submission does not contain these data, please either upload them as Supporting Information files or deposit them to a stable, public repository and provide us with the relevant URLs, DOIs, or accession numbers. For a list of recommended repositories, please see https://journals.plos.org/plosone/s/recommended-repositories. If there are ethical or legal restrictions on sharing a de-identified data set, please explain them in detail (e.g., data contain potentially sensitive information, data are owned by a third-party organization, etc.) and who has imposed them (e.g., an ethics committee). Please also provide contact information for a data access committee, ethics committee, or other institutional body to which data requests may be sent. If data are owned by a third party, please indicate how others may request data access. 6. PLOS ONE now requires that authors provide the original uncropped and unadjusted images underlying all blot or gel results reported in a submission’s figures or Supporting Information files. This policy and the journal’s other requirements for blot/gel reporting and figure preparation are described in detail at https://journals.plos.org/plosone/s/figures#loc-blot-and-gel-reporting-requirements and https://journals.plos.org/plosone/s/figures#loc-preparing-figures-from-image-files. When you submit your revised manuscript, please ensure that your figures adhere fully to these guidelines and provide the original underlying images for all blot or gel data reported in your submission. See the following link for instructions on providing the original image data: https://journals.plos.org/plosone/s/figures#loc-original-images-for-blots-and-gels.   In your cover letter, please note whether your blot/gel image data are in Supporting Information or posted at a public data repository, provide the repository URL if relevant, and provide specific details as to which raw blot/gel images, if any, are not available. Email us at plosone@plos.org if you have any questions.

Additional Editor Comments:

Thank you for submitting the following manuscript to PLOS ONE.

Please revise the manuscript according to the reviewers' comments and upload the revised file.

Reviewers' comments:

Reviewer's Responses to Questions

**Comments to the Author**

1. Is the manuscript technically sound, and do the data support the conclusions?

Reviewer #1: Yes

Reviewer #2: No

2. Has the statistical analysis been performed appropriately and rigorously? 

Reviewer #1: Yes

Reviewer #2: Yes

3. Have the authors made all data underlying the findings in their manuscript fully available?

Reviewer #1: Yes

Reviewer #2: Yes

4. Is the manuscript presented in an intelligible fashion and written in standard English?

Reviewer #1: Yes

Reviewer #2: Yes

5. Review Comments to the Author

Reviewer #1: I believe the authors have done a good job in the conduct and report of their experimental findings. A few concerns however. The Figure 1a legend mentions data represents "MEAN" of 4 duplicated experiments. I understand this to mean four independent experiments each with duplicate wells. However, Fig1a has more data points for control and IBMX compared to the inhibitor-treated groups. Can the authors provide an explanation for this?

Furthermore, the effect of H2O2 on PAI activity and protein expression (Fig 7a and B), used as a model of endothelial cell pathology does not appear to show statistical significance when compared to control. Thus, the data does not show, at least in this study, that endothelial cell dysfunction occurred as measured by PAI activity or protein expression. It would therefore be difficult to argue that COMP4 protects against endothelial dysfunction. The authors are advised to revise this conclusion.

Reviewer #2: The manuscript by Ferrini and colleagues is investigating the NO effects of the Nutraceutical COMP-4 in culture. The authors used different NO measurements tools, and cytokine bead arrays to show that COMP-4 treatment induced cGMP and broad of inhibitors to NOS decreased this effect. This was associated with the same effect on NO endogenous levels. Western blot and qRT-PCR analysis showed that eNOS and iNOS induced with COMP4, and there was a decrease in PAI1 and IL-8. While H2O2 pretreatment decreased nitrite levels, COMP-4 prevented that, same with arrays of cytokines. Overall, this work is interesting and might provide a mechanistic insight on the COMP-4 on NO, especially there is a clinical study is going on in UCLA. The authors have similar observations with COMP4 on SMCs and the same pathway was characterized in their work that was published in 2021 (PMID: 34430391). This brought the novelty of the work down. Additional major and minor concerns are listed below:

Major concerns:

1. The reason why this cell line was picked for the study?

2. What does that mean "both inhibitors decreased cGMP" in figure 1A? also, should nt these inhibitors given with IBMX? what each dot in the graphs means? individual data, repeats? especially in the figure legends stated the data are from n=4, but obviously there are more than 4 dots, and there are as less as 2 dots in figure 1B.

3. Figure 2 should be done with IBMX+/- inhibitors for the consistency of the data.

4. Figure 3, COMP4 seems to decrease nNOS?

5. No differences between H2O2 treatment and control, should be different dosing use to test?

6. Figure 7, please show the bands without the boxes and with loading controls, and the quantification should be relative to loading control.

Minor concerns:

1. Passage number is missing for the cells in culture.

2. Final concentration of COMP-4 and what is the final % of ethanol used in culture.

3. Methods: sources of all inhibitors used, cata. No, and final doses and why?

4. Citations should be consistent, numbers or names/years throughout the text.

5. sources of H2O2, especially the effect in the experiments is mild and it does not seem to induce anything from the graphs.

6. PLOS authors have the option to publish the peer review history of their article (what does this mean?). If published, this will include your full peer review and any attached files.

Reviewer #1: No

Reviewer #2: **Yes: **Mabruka Alfaidi

---

## [Author Response · Author response to Decision Letter 0]

12 Nov 2024

RESPONSE TO REVIEWER COMMENTS

Dear Editor:

We appreciate the constructive feedback offered by the reviewers. We have addressed each reviewer comment below, and we have updated the manuscript to reflect all described changes. 

We have also updated the manuscript to include all required formatting advised in the PLOS ONE decision email, an updated Competing Interests statement, and inclusion of supplementary figures/tables to conform with PLOS ONE’s Data Availability requirements. The original underlying images for all blot and gel data are included in Supporting Information files. 

As requested, here is our updated Competing Interests statement: We have read the journal’s policy, and the authors of this manuscript have the following competing interests: JR is a principal in KLRM, LLC, which holds the patent for COMP-4. All other authors have declared that no competing interests exist. This does not alter our adherence to PLoS ONE policies on sharing data and materials.

---- 

COMMENTS FROM REVIEWER 1:

I believe the authors have done a good job in the conduct and report of their experimental findings. A few concerns however.

Comment 1: The Figure 1a legend mentions data represents "MEAN" of 4 duplicated experiments. I understand this to mean four independent experiments each with duplicate wells. However, Fig1a has more data points for control and IBMX compared to the inhibitor-treated groups. Can the authors provide an explanation for this?

Response: We thank the reviewer for their comments and suggestions for improving the manuscript. The inhibitor experiments (L-NAME and L-NIL) were conducted separately, but Control, IBMX, and COMP-4 were run consistently to normalize values. Each point represents the average of duplicate wells from two or four independent experiments, as indicated. We will update the figure legend for clarity.

Manuscript change: The new Figure 1A and 1B legends will read: “Results for cGMP quantitation are expressed as pmol/mg protein and represent the mean ± S.E.M. of two and four experiments done in duplicates. Statistical differences were examined by one-way ANOVA followed by Tukey multiple comparison test. **p<0.01; ***p<0.001 vs. control; #p<0.05; ## p<0.01 with respect to COMP-4.”

Comment 2: Furthermore, the effect of H2O2 on PAI activity and protein expression (Fig 7A and B), used as a model of endothelial cell pathology, does not appear to show statistical significance when compared to control. Thus, the data does not show, at least in this study, that endothelial cell dysfunction occurred as measured by PAI activity or protein expression. It would therefore be difficult to argue that COMP4 protects against endothelial dysfunction. The authors are advised to revise this conclusion.

Response: We acknowledge the reviewer's observation that the effect shown in Figure 7 was not as significant as in the cytokine experiments and nitrite production. Our primary focus was employing hydrogen peroxide as a reactive oxygen species (ROS) agent to examine how COMP-4 prevents damage. The H2O2 concentration was kept sublethal to avoid cell death and observe functional impairment. Although the effect on PAI-1 was not statistically significant, there was a trend toward increased expression with H2O2.

Manuscript change: We have revised the manuscript to reflect "endothelial function impairment" rather than the term "endothelial dysfunction.” We have made this change in the Introduction, Methods, Results, and Discussion.

COMMENTS FROM REVIEWER 2:

The manuscript by Ferrini and colleagues is investigating the NO effects of the Nutraceutical COMP-4 in culture. The authors used different NO measurements tools, and cytokine bead arrays to show that COMP-4 treatment induced cGMP and broad of inhibitors to NOS decreased this effect. This was associated with the same effect on NO endogenous levels. Western blot and qRT-PCR analysis showed that eNOS and iNOS induced with COMP4, and there was a decrease in PAI1 and IL-8. While H2O2 pretreatment decreased nitrite levels, COMP-4 prevented that, same with arrays of cytokines. Overall, this work is interesting and might provide a mechanistic insight on the COMP-4 on NO, especially there is a clinical study is going on in UCLA. The authors have similar observations with COMP4 on SMCs and the same pathway was characterized in their work that was published in 2021 (PMID: 34430391). This brought the novelty of the work down. Additional major and minor concerns are listed below.

Major Comment 1 (paraphrasing): Why was this cell line (HUAEC) chosen?

Response: We thank the reviewer for asking the question. HUAEC were chosen due to their similarity to systemic arterial endothelial cells. This is supported by a DNA microarray-based study which showed that HUAEC cluster with endothelial cells from arteries in the systemic circulation (in contrast with HUVEC, which cluster with endothelial cells from other systemic veins) (Chi et al., PNAS 2003, PMID 12963823). 

Manuscript change: We have updated the Materials and Methods section to reflect the motivation for selecting HUAEC for this study, and add the reference mentioned above. The text now reads: “The HUAEC used in our experiments were selected based on previously published work using DNA microarrays, which showed that these cells have similar characteristics to systemic arterial endothelial cells.”

Major Comment 2: What do you mean by “both inhibitors decreased cGMP” in Figure 1A? Also, shouldn’t these inhibitors be given with IBMX? What does each dot in the graphs mean? Individual data, repeats? The figure legends state that the data are from n=4, but obviously, there are more than 4 dots, and there are as few as 2 dots in Figure 1B.

Response: We appreciate the reviewer’s attention to these details and would like to clarify the following points:

1) In Figure 1A, we showed that incubating HUAEC with the non-specific NOS inhibitor L-NAME resulted in reduction in cGMP expression, but incubating HUAEC with the selective iNOS inhibitor L-NIL resulted in no inhibition of cGMP expression. In Figure 1B, we show that COMP-4 increases cGMP expression, while co-incubation of COMP-4 with either L-NAME or L-NIL reduces cGMP expression, suggesting a mechanism that includes iNOS. 

2) We excluded IBMX from the inhibitor experiments because IBMX, as a non-specific phosphodiesterase inhibitor, acts downstream of the NO-cGMP pathway. Including IBMX in the inhibitor experiments would have confounded our ability to isolate the upstream effects of NOS inhibition on cGMP production, which was the primary focus of these experiments, and co-incubation of IBMX with NOS inhibitors would not add new information related to the effect of COMP-4 in this cell line.

3) In Figures 1A and 1B, we ran L-NIL and L-NAME in separate experiments but always together with Control, IBMX, and COMP-4. We combined all the points to create the figures. The experiment with L-NAME was conducted four times, with duplicate wells each time, and each point on the figures represents the average of two independent values. The experiment with L-NIL was conducted twice. This is the discrepancy in the number of points in the figure. 

4) The figure legend states that the data are from n=4, but this refers to the number of independent experimental repeats. For some conditions (e.g., control and IBMX), additional experiments were conducted to normalize the data. Conversely, for L-NIL in Figure 1B, fewer experiments (n=2) were performed, leading to fewer data points. The legend has been updated to clarify this point.

Manuscript change: The corrected legends for Figure 1 now read: Results for cGMP quantitation are expressed as pmol/mg protein and represent the mean ± S.E.M. of two and four experiments done in duplicates. Statistical differences were examined by one-way ANOVA followed by Tukey multiple comparison test. **p<0.01; ***p<0.001 vs. control; #p<0.05; ## p<0.01 with respect to COMP-4.

Major Comment 3: Figure 2 should be done with IBMX+/- inhibitors for the consistency of the data.

Response: We appreciate the reviewer's suggestion to include IBMX in Figure 2, but respectfully disagree. Our primary objective in Figure 2 was to measure the intracellular nitric oxide (NO) production in response to COMP-4 treatment, rather than the downstream effects on cGMP. Since IBMX is a phosphodiesterase inhibitor that prevents cGMP degradation, including it would not directly impact NO production, which was the focus of these experiments. Inhibitors like L-NAME and L-NIL specifically target the upstream nitric oxide synthase (NOS) enzymes that regulate NO synthesis, making them the most relevant compounds to test in this context. For consistency, we used IBMX in experiments involving cGMP measurement (e.g., Figure 1), where its role in preventing cGMP degradation is directly relevant. In contrast, Figure 2 is designed to isolate the effects of COMP-4 on NO production, independent of cGMP degradation. Therefore, we did not include IBMX in these experiments.

Manuscript change: We have added the clarification: “Since the primary objective was to examine intracellular NO production in response to COMP-4 treatment, independent of downstream effects on cGMP degradation, IBMX was not included in the experiments described in Fig 2.”

Major Comment 4: Figure 3, COMP-4 seems to decrease nNOS?

Response: We thank the reviewer for pointing this out. The observed decrease in nNOS expression was minor and not statistically significant. Given that baseline nNOS levels in these cells were very low, we focused primarily on the significant increases observed for eNOS and iNOS. The observation that nNOS is minimally responsive to COMP-4 has been reported previously in our work with other cell lines (e.g., penile corporal smooth muscle cells).

Manuscript change: We have added a statement in the "Results" section indicating that the decrease in nNOS expression was not statistically significant (and therefore did not contribute meaningfully to the study’s outcomes).

Major Comment 5: No differences between H2O2 treatment and control; should there be different dosing used to test?

Response: We acknowledge the reviewer’s observation that the comparison between H2O2 and Control was not significant. Our primary focus was employing hydrogen peroxide as a reactive oxygen species (ROS) agent to examine how COMP-4 prevents injury. The H2O2 concentration was kept sublethal to avoid cell death and observe functional impairment. Although the effect on PAI-1 was not statistically significant, there was a trend toward increased expression with H2O2.

Major Comment 6: In Figure 7, please show the bands without the boxes and with loading controls, and the quantification should be relative to the loading control.

Response: We have removed the boxes on the blot images. The quantification was done using the new Lycor photo-documentation system, which uses total protein normalization instead of loading control. It has been shown that total protein normalization (TPN) is more accurate than housekeeping proteins (HKP). HKP expression is more variable than constant, changing with cell type and developmental stage. We have included the TPN for each blot as an example.

Manuscript change: We have removed the boxes on the graphs. We clarified the use of total protein normalization (TPN) in the Materials and Methods section.

Minor comment 1: Passage number is missing for the cells in culture.

Response: We used HUAEC between passages 2 through 5 because cellular functions tend to be most consistent within this range.

Manuscript change: The Materials and Methods section has been updated to reflect our use of cells between passages 2 through 5.

Minor comment 2: Final concentration of COMP-4 and what is the final % of ethanol used in culture.

Response: We acknowledge the reviewer’s point that we did not provide sufficient details regarding the amount of ethanol and the final concentration of COMP-4 after mixing each component. The final concentration after mixing all the components in 0.69 mg/ml. This is adapted from reference 20 (Ferrini et al., Nitric Oxide 2018).

In our experiments we initially prepared stock solutions at a 100-fold concentration for MP, PC, and L-citrulline (at 90 mg/ml) and at a 200-fold concentration for ginger (at 40 mg/ml). All stock solutions were prepared in 70% ethanol, except for L-citrulline, which was prepared in water due to its solubility. We added 10 μl of the components per 1 ml of media, resulting in a 0.07% ethanol concentration received by both the control and treated cells, which is not considered lethal for the cells. Furthermore, in all the experiments conducted in this study, the control cells received the same amount of ethanol as the treated cells.

In our previous studies, in order to ensure that the levels of cGMP were not affected by the addition of ethanol, we measured the levels of cGMP in untreated (control) cells with and without the addition of ethanol, and no significant differences were observed in either group. 

Manuscript change: We have added the following text to the Materials and Methods section:

COMP-4 is a mixture of four components created by combining PC, MP, L-citrulline, and ginger. The final concentration of the mixture is 0.69 mg/ml. PC, MP, and L-citrulline were prepared at a 100-fold stock solution, while ginger was prepared at a 200-fold stock solution due to solubility reasons. All components were dissolved in 70% ethanol, except for L-citrulline, which was dissolved in water.

In the experiments, 10 μl of the COMP-4 mixture was added per ml of media, representing an addition of 0.07% of ethanol. This concentration of ethanol is not considered lethal for the cells. The control cells received the same amount of ethanol (0.07%) as the treated cells in all the experiments. 

Minor comment 3: Methods: sources of all inhibitors used, catalog number, and final doses and why?

Response: All the sources of the inhibitors and catalog numbers have been added.

Manuscript change: All the sources of the inhibitors and catalog numbers have been added.

Minor comment 4: Citations should be consistent, with numbers or names/years throughout the text.

Response: We thank the reviewers for their attention to this. We will make these changes and modify according to journal guidelines. 

Manuscript change: Citations have been updated according to journal guidelines and verified. 

Minor comment 5: Sources of H2O2, especially the effect in the experiments, are mild, and it does not seem to induce anything from the graphs.

Response: We have added the source of the H2O2 reagent. We appreciate the reviewer’s observation regarding the mild effects of H2O2 in our experiments. We intentionally used a sublethal concentration of H2O2 (100 µM) to avoid inducing cell death and instead focus on oxidative stress-induced functional impairments in endothelial cells. This concentration was chosen based on prior studies that demonstrated its ability to induce oxidative stress without overwhelming the cells, allowing us to observe subtle but important changes in nitric oxide (NO) production, cytokine expression, and PAI-1 activity. While the effects of H2O2 may appear mild in some graphs, we observed consistent trends indicating increased oxidative stress, such as the upregulation of pro-inflammatory cytokines (e.g., IL-6, IL-8) and PAI-1 expression. These changes were mitigated by co-treatment with COMP-4, suggesting that COMP-4 offers protection against oxidative stress-induced impairments.

Manuscript change: We have revised both the Materials and Methods and the Discussion sections to emphasize that the sublethal concentration of H2O2 was specifically chosen to impair endothelial function without causing significant cell death, and that the protective effects of COMP-4 were observed under these conditions.

----

We appreciate the totality of the reviewers’ comments on our manuscript. We feel that the comments, suggestions, and inquiries have meaningfully strengthened our paper. We thank the reviewers for taking the time to provide their input.

Sincerely,

Monica Ferrini, Andrea Abraham, Revecca Millán, Leslie 

---

## [Decision Letter · Decision Letter 1]

18 Dec 2024

Nutraceutical COMP-4 confers protection against endothelial dysfunction through the eNOS/iNOS-NO-cGMP pathway

PONE-D-24-27761R1

Dear Dr. Eleswarapu,

We’re pleased to inform you that your manuscript has been judged scientifically suitable for publication and will be formally accepted for publication once it meets all outstanding technical requirements.

Kind regards,

Yung-Hsiang Chen, Ph.D.

Academic Editor

PLOS ONE

Additional Editor Comments (optional):

Congratulations on the acceptance of your manuscript, and thank you for your interest in submitting your work to PLOS ONE.

Reviewers' comments:

Reviewer's Responses to Questions

**Comments to the Author**

1. If the authors have adequately addressed your comments raised in a previous round of review and you feel that this manuscript is now acceptable for publication, you may indicate that here to bypass the “Comments to the Author” section, enter your conflict of interest statement in the “Confidential to Editor” section, and submit your "Accept" recommendation.

Reviewer #3: All comments have been addressed

2. Is the manuscript technically sound, and do the data support the conclusions?

Reviewer #3: Yes

3. Has the statistical analysis been performed appropriately and rigorously? 

Reviewer #3: Yes

4. Have the authors made all data underlying the findings in their manuscript fully available?

Reviewer #3: Yes

5. Is the manuscript presented in an intelligible fashion and written in standard English?

Reviewer #3: Yes

6. Review Comments to the Author

Reviewer #3: The study investigates the protective effects of the nutraceutical COMP-4 on endothelial dysfunction. COMP-4, which consists of L-citrulline, ginger extract, Paullinia cupana, and muira puama, was tested on human umbilical arterial endothelial cells (HUAEC). The researchers found that COMP-4 increases the production of nitric oxide (NO) and cyclic guanosine monophosphate (cGMP) by upregulating endothelial nitric oxide synthase (eNOS) and inducible nitric oxide synthase (iNOS). This upregulation leads to a reduction in pro-inflammatory cytokines and plasminogen activator inhibitor-1 (PAI-1) activity. Additionally, COMP-4 mitigates the detrimental effects of hydrogen peroxide (H2O2) on endothelial cells, suggesting its potential role in protecting against oxidative stress-induced endothelial dysfunction.

The revised manuscript addresses the reviewers' comments comprehensively and provides a clear and detailed account of the experimental findings. The study's methodology is robust, and the results are well-presented and statistically sound. The authors have successfully demonstrated the protective effects of COMP-4 on endothelial function, highlighting its potential therapeutic benefits. Given the thorough revisions and the significance of the findings, I believe this manuscript is suitable for publication in its current form.

7. PLOS authors have the option to publish the peer review history of their article (what does this mean?). If published, this will include your full peer review and any attached files.

Reviewer #3: No

---

## [Editor Report · Acceptance letter]

22 Dec 2024

PONE-D-24-27761R1 

PLOS ONE

Dear Dr. Eleswarapu, 

I'm pleased to inform you that your manuscript has been deemed suitable for publication in PLOS ONE. Congratulations! Your manuscript is now being handed over to our production team.

Kind regards, 

on behalf of

Dr. Yung-Hsiang Chen 

Academic Editor

PLOS ONE
